# Mitochondrial inner membrane permeabilisation enables mtDNA release during apoptosis

Joel S Riley[1,2] (ID), Giovanni Quarato[3] (ID), Catherine Cloix[1,2], Jonathan Lopez[1,2,†] (ID), Jim O'Prey[1,2], Matthew Pearson[4], James Chapman[5], Hiromi Sesaki[6], Leo M Carlin[1,2] (ID), João F Passos[5,7], Ann P Wheeler[4], Andrew Oberst[8,9], Kevin M Ryan[1,2] & Stephen WG Tait[1,2,*] (ID)

## Abstract

During apoptosis, pro-apoptotic BAX and BAK are activated, causing mitochondrial outer membrane permeabilisation (MOMP), caspase activation and cell death. However, even in the absence of caspase activity, cells usually die following MOMP. Such caspase-independent cell death is accompanied by inflammation that requires mitochondrial DNA (mtDNA) activation of cGAS-STING signalling. Because the mitochondrial inner membrane is thought to remain intact during apoptosis, we sought to address how matrix mtDNA could activate the cytosolic cGAS-STING signalling pathway. Using super-resolution imaging, we show that mtDNA is efficiently released from mitochondria following MOMP. In a temporal manner, we find that following MOMP, BAX/BAK-mediated mitochondrial outer membrane pores gradually widen. This allows extrusion of the mitochondrial inner membrane into the cytosol whereupon it permeablises allowing mtDNA release. Our data demonstrate that mitochondrial inner membrane permeabilisation (MIMP) can occur during cell death following BAX/BAK-dependent MOMP. Importantly, by enabling the cytosolic release of mtDNA, inner membrane permeabilisation underpins the immunogenic effects of caspase-independent cell death.

**Keywords** apoptosis; BAX/BAK; cGAS-STING; mitochondria; mtDNA
**Subject Categories** Autophagy & Cell Death; Membrane & Intracellular Transport; Signal Transduction
**The EMBO Journal (2018) 37: e99238**

See also: **K Cosentino & AJ García-Sáez** (September 2018)

## Introduction

To initiate cell death, intrinsic or mitochondrial apoptosis requires mitochondrial outer membrane permeabilisation, or MOMP (Tait & Green, 2013). MOMP causes the release of mitochondrial intermembrane space proteins, including cytochrome *c*, that activate caspase proteases leading to apoptosis. Nevertheless, irrespective of caspase activity, cells typically do not survive following widespread MOMP, defining it as a point-of-no-return (Haraguchi *et al*, 2000; Ekert *et al*, 2004; Tait *et al*, 2010). Because of this central role in dictating cell fate, MOMP is tightly controlled, primarily via pro- and anti-apoptotic members of the BCL-2 protein family (Lopez & Tait, 2015). Under homeostatic conditions, anti-apoptotic BCL-2 family members (e.g. BCL-2, BCL-xL, MCL-1) block the pro-apoptotic actions of BAX and BAK. Following an apoptotic trigger, BAX and BAK are activated, leading to their oligomerisation in the mitochondrial outer membrane and MOMP (Wei *et al*, 2001; Cosentino & Garcia-Saez, 2017).

Mitochondrial apoptosis is considered a non-inflammatory form of cell death, allowing the host to quickly and efficiently clear away dead cell corpses without provoking an immune response (Arandjelovic & Ravichandran, 2015). Recently, others and ourselves have shown that caspase activity is essential for the non-inflammatory nature of mitochondrial apoptosis; if caspase activity is blocked following MOMP, cell death still occurs, but a type I interferon (IFN) response and NF-κB activation ensues (Rongvaux *et al*, 2014; White *et al*, 2014; Giampazolias *et al*, 2017). This leads to pro-inflammatory cytokine production and an immune response towards the dying cell that can initiate anti-tumour immunity (Giampazolias *et al*, 2017). Therefore, as proposed by others, a main function of apoptotic caspase activity may be to silence inflammation during cell death (Martin *et al*, 2012).

1   Cancer Research UK Beatson Institute, Glasgow, UK
2   Institute of Cancer Sciences, University of Glasgow, Glasgow, UK
3   Department of Immunology, St. Jude Children's Research Hospital, Memphis, TN, USA
4   MRC Human Genetics Unit, MRC Institute of Genetics and Molecular Medicine, The University of Edinburgh, Edinburgh, UK
5   Ageing Research Laboratories, Newcastle University Institute for Ageing, LLHW Centre for Ageing and Vitality, Campus for Ageing and Vitality, Newcastle University, Newcastle upon Tyne, UK
6   Department of Cell Biology, Johns Hopkins University School of Medicine, Baltimore, MD, USA
7   Institute for Cell and Molecular Biosciences, Newcastle University, Newcastle upon Tyne, UK
8   Department of Immunology, University of Washington, Seattle, WA, USA
9   Center for Innate Immunity and Immune Disease, University of Washington, Seattle, WA, USA
    *Corresponding author. Tel: +44 1413 308703; E-mail: stephen.tait@glasgow.ac.uk
    †Present address: UMR INSERM 1052 CNRS 5286, Cancer Research Centre of Lyon (CRCL), Léon Bérard Centre, University of Lyon, Lyon, France

The ability of MOMP to activate inflammation and a type I IFN response requires recognition of mtDNA by the cytosolic cGAS-STING DNA sensing pathway (Rongvaux *et al*, 2014; White *et al*, 2014; Giampazolias *et al*, 2017). However, it is challenging to reconcile how matrix localised mtDNA can activate the cytosolic cGAS-STING DNA sensing pathway since the mitochondrial inner membrane is thought to remain intact during apoptosis (von Ahsen *et al*, 2000; Waterhouse *et al*, 2001). Given this paradox, we set out to define how mtDNA could activate cGAS-STING signalling. Using a variety of high-resolution imaging techniques, we find that following the onset of MOMP, BAX/BAK-mediated pores widen to allow the extrusion of newly unstructured inner membrane. Under conditions of caspase inhibition, the extruded inner membrane permeabilises facilitating mtDNA release into the cytoplasm, allowing it to activate cGAS-STING signalling and IFN synthesis. Unexpectedly, our data demonstrate that the mitochondrial inner membrane can undergo permeabilisation during cell death. Importantly, by releasing mtDNA, mitochondrial inner membrane permeabilisation, or MIMP, enables cell death-associated inflammation.

## Results

### mtDNA is released from mitochondria following MOMP

To visualise mtDNA dynamics during apoptosis, we used super-resolution Airyscan confocal microscopy. Importantly, this provides sufficient resolution to allow the visualisation of the mitochondrial outer membrane and matrix-resident molecules (such as TFAM containing mtDNA nucleoids) as distinct entities under sub-diffraction-limiting conditions. As expected, in healthy U2OS cells, dual immunostaining of the mitochondria outer membrane protein TOM20 and DNA revealed that mtDNA nucleoids are contained within mitochondria surrounded by a continuous outer membrane (Fig 1A). To engage mitochondrial apoptosis, U2OS cells were treated with the BH3-mimetic ABT-737, which inhibits BCL-xL, BCL-2 and BCL-w (Oltersdorf *et al*, 2005) together with actinomycin D (ActD, to inhibit transcription of labile MCL-1), in the presence or absence of pan-caspase inhibitor qVD-OPh. To assess cell viability, U2OS cells were imaged using IncuCyte live-cell imaging and SYTOX Green uptake or assessed for long-term clonogenic survival (Fig EV1A and B). Consistent with engagement of mitochondrial apoptosis, ABT-737/ActD co-treatment rapidly and synchronously induced cell death (Fig EV1A). Co-treatment with caspase inhibitor qVD-OPh prevented cell death in the short term (Fig EV1A) but failed to allow long-term clonogenic survival (Fig EV1B), consistent with widespread MOMP being a point-of-no-return. Under identical MOMP-inducing conditions, U2OS cells were immunostained with anti-TOM20 and DNA antibodies to visualise mitochondrial outer membrane integrity and mtDNA-containing nucleoids, respectively (Fig 1B). Alternatively, U2OS cells were immunostained with anti-TOM20 and cytochrome *c* antibodies to determine MOMP (Fig EV1C and D). Under these conditions, over 90% of U2OS cells underwent MOMP, as determined by loss of mitochondrial cytochrome *c* staining (Fig EV1C and D). Specifically following ABT-737/ActD treatment, under caspase-inhibited conditions, permeabilisation of the mitochondrial outer membrane was observed in over 80% of cells, as evidenced by

discontinuous, crescent-like TOM20 immunostaining (Fig 1B and C). Strikingly, mtDNA displayed cytosolic re-localisation in cells that had undergone MOMP under caspase-inhibited conditions (Fig 1B). Under these conditions, over 80% of cells displayed mtDNA cytosolic release (Fig 1C) and, on average per cell, over 80% of mtDNA displayed cytosolic localisation (Fig 1D). A similar pattern of mtDNA cytosolic localisation and mitochondrial outer membrane permeabilisation was also observed in E1A/Ras transformed MEF specifically following ABT-737/ActD/qVD-OPh treatment (Fig 1E). Both cytosolic and mitochondrial mtDNA structures were of similar size, suggesting cytosolic re-localisation of mtDNA-containing nucleoids. To determine whether mtDNA re-localisation was dependent on MOMP, we used CRISPR-Cas9 genome editing to delete BAX and BAK, two proteins essential for MOMP (Wei *et al*, 2001) either singly or together (Fig 1F). Only combined deletion of BAX and BAK in U2OS cells prevented MOMP and cell death following ABT-737/S63845 treatment (Figs 1G and EV1E). Under identical conditions, dual immunostaining of TOM20 and DNA demonstrated that combined deletion of BAX/BAK, but not of either alone, prevented cytosolic release of mtDNA following ABT-737/S63845/qVD-OPh treatment (Fig 1H and I). Collectively, these data show that matrix localised mtDNA is released from mitochondria following BAX/BAK-dependent MOMP under caspase-inhibited conditions.

### Mitochondrial release of mtDNA correlates with a STING-dependent interferon response

We next investigated an association between mitochondrial mtDNA release and activation of a STING-dependent interferon response. For this purpose, we used SVEC 4–10 murine endothelial cells, which we have previously shown have an intact cGAS-STING signalling pathway (Giampazolias *et al*, 2017). To induce rapid mitochondrial apoptosis, SVEC cells were treated with ABT-737 together with the MCL-1 inhibitor S63845 (Kotschy *et al*, 2016), and cell death was monitored by SYTOX Green uptake and IncuCyte live-cell imaging (Fig 2A). Combined ABT-737/S63845 treatment led to rapid cell death that was inhibited by addition of qVD-OPh, consistent with engagement of mitochondrial apoptosis. Following 3 h of ABT-737/S63845 treatment in the presence of qVD-OPh, cells were immunostained with anti-TOM20 and DNA antibodies to visualise mitochondrial outer membrane integrity and mtDNA-containing nucleoids (Fig 2B). Similar to other cell types, mitochondrial release of mtDNA was detected. We next investigated whether a STING-dependent transcriptional interferon response was being activated under these conditions using qRT–PCR. Following engagement of caspase-independent cell death (CICD), various IFN response genes (*Ifnb1*, *Irf7* and *Oasl1*) were upregulated (Fig 2C). ABT-737/S63845-induced transcriptional upregulation of *Ifnb1* was greatly increased upon inhibition of caspase function by qVD-OPh addition (Fig 2D). Deletion of STING, through CRISPR/Cas9 genome editing (Fig 2E), blocked *Ifnb1* upregulation during CICD, consistent with previous findings of others and ourselves (Fig 2F; Giampazolias *et al*, 2017; Rongvaux *et al*, 2014; White *et al*, 2014). Combined deletion of BAX and BAK, but not of either alone, also inhibited *Ifnb1* upregulation following ABT-737/S63845/qVD-OPh treatment, demonstrating a requirement for MOMP (Fig 2G and H). These data

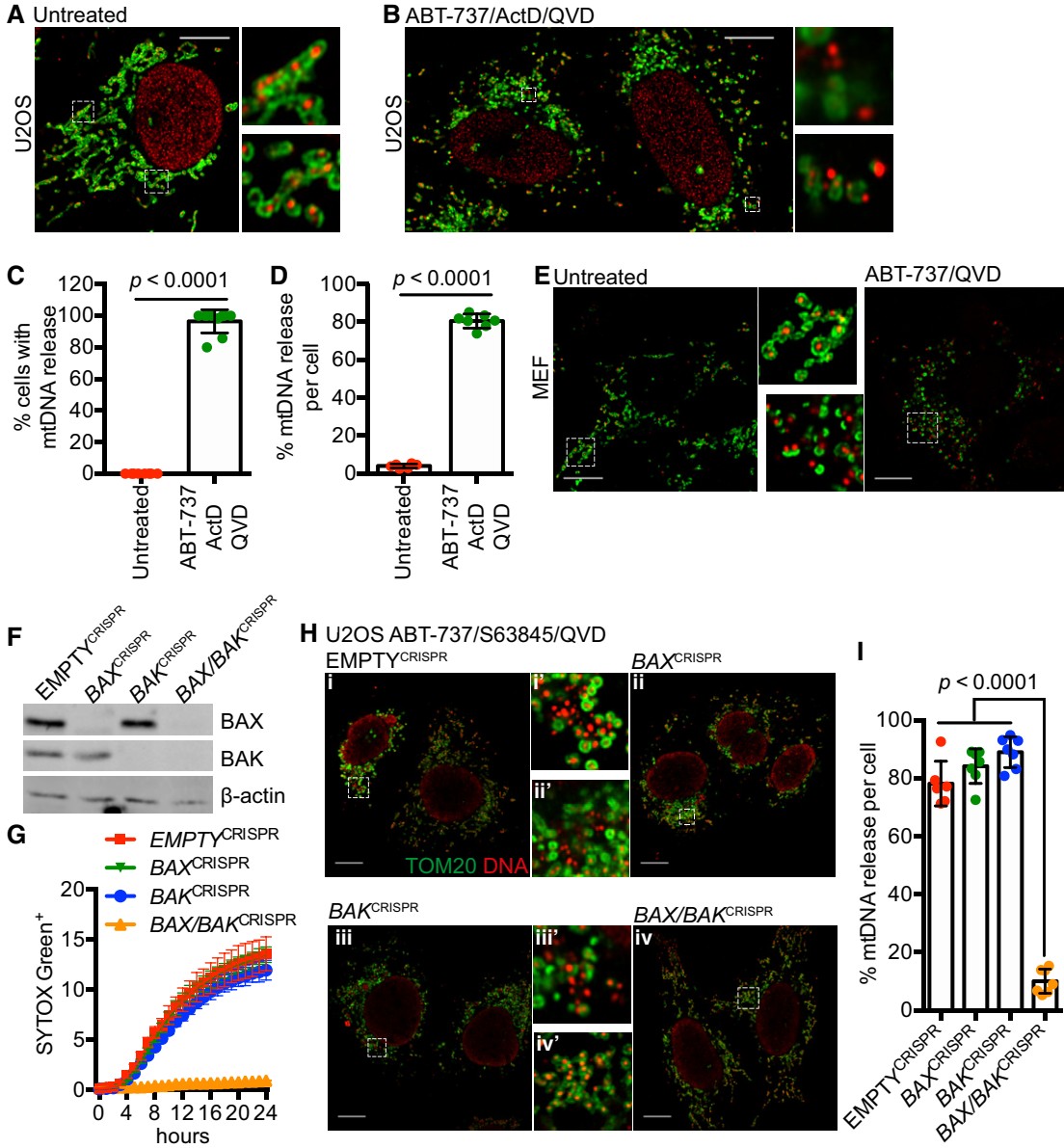

**Figure 1.  mtDNA is released from mitochondria following MOMP in a BAX/BAK-dependent manner.**

A   Fixed super-resolution Airyscan images of U2OS cells immunostained with anti-TOM20 (green) and anti-DNA (red) antibodies. Scale bar = 10 μm. Representative images from three independent experiments.

B   Airyscan images of U2OS cells treated with 10 μM ABT-737, 1 μM ActD and 20 μM qVD-OPh for 3 h, immunostained with anti-TOM20 and anti-DNA antibodies. Scale bar = 10 μm. Representative images from three independent experiments.

C   Quantification of cells exhibiting > 10% mtDNA release following treatment with 10 μM ABT-737, 1 μM ActD and 20 μM qVD-OPh. Data are expressed as mean ± SD from three independent experiments and analysed by Student's *t*-test.

D   Quantification of the extent of mitochondrial DNA (mtDNA) nucleoid release per cell following treatment with 10 μM ABT-737, 1 μM ActD and 20 μM qVD-OPh. Data are expressed as mean ± SD from two independent experiments and analysed by Student's *t*-test.

E   Airyscan images of MEF cells untreated or treated with 10 μM ABT-737 and 20 μM qVD-OPh for 3 h, immunostained with anti-TOM20 and anti-DNA antibodies. Scale bar = 10 μm. Representative images from three independent experiments.

F   BAX and BAK expression levels in U2OS cells with CRISPR-Cas9-mediated deletion of BAX, BAK or BAX/BAK.

G   U2OS cells with BAX, BAK or BAX/BAK deletion by CRISPR-Cas9 treated with 10 μM ABT-737 and 2 μM S63845 and analysed for cell viability using an IncuCyte live-cell imager and SYTOX Green exclusion. Data are expressed as mean ± SEM, representative of three independent experiments, and have been normalised to starting confluency.

H   Airyscan images of U2OS (i) control cells or with CRISPR-Cas9-mediated deletion of either (ii) BAX, (iii) BAK or (iv) BAX and BAK treated with 10 μM ABT-737, 2 μM S63845 and 20 μM qVD-OPh for 3 h. Scale bar = 10 μm. Representative images from three independent experiments.

I   Quantification of mtDNA nucleoid release per cell in U2OS EMPTY[CRISPR], *BAX*[CRISPR], *BAK*[CRISPR] and *BAX/BAK*[CRISPR] cells. Data are expressed as mean ± SD from three independent experiments and analysed using Student's *t*-test.

Source data are available online for this figure.

demonstrate a correlation between the release of mtDNA and activation of a STING-dependent interferon response.

## Mitochondrial nucleoids are extruded via expanding outer membrane pores

Our data suggests that mtDNA nucleoids are released from mitochondria during following MOMP under caspase-inhibited conditions. To investigate this further, U2OS cells were treated with ABT-737/S63845/qVD-OPh for 3 h to induce CICD, then simultaneously stained for mtDNA and TFAM—a nucleoid-resident protein (Fig 3A). The mitochondrial outer membrane was visualised using a SNAP-tag Omp25 fusion protein and the far-red fluorophore Janelia Fluor 646 ($JF_{646}$; referred to as $JF_{646}$-MOM; Grimm *et al*, 2015; Katajisto *et al*, 2015). As expected, in healthy mitochondria, DNA co-localised with TFAM within the mitochondria (Fig 3A). Importantly, during CICD, both DNA and TFAM co-localised outside the mitochondria, consistent with release of mtDNA-containing nucleoids. We next investigated the dynamics of mtDNA re-localisation by live-cell imaging. Because live-cell imaging of mtDNA using Pico-Green was not possible due to photobleaching, we monitored the localisation of mClover-fused TFAM. The mitochondrial outer membrane was visualised using $JF_{646}$-MOM. As expected, TFAM-mScarlet localised to the mitochondrial matrix, co-localising with mtDNA (Fig EV2A). U2OS cells expressing $JF_{646}$-MOM and TFAM-mClover were induced to undergo MOMP ± qVD-OPh and imaged by live-cell imaging (Fig EV2B, Videos EV1–EV3). Under caspase-proficient conditions, following MOMP, the cell rapidly shrunk and detached, consistent with engagement of apoptosis (Video EV1). Under caspase-inhibited conditions, after ABT-737 treatment, TFAM-mClover was released into the cytosol, consistent with our earlier analysis of mtDNA (Fig EV2B, Videos EV2 and EV3). Finally, we investigated the kinetics of TFAM-mClover release and MOM pore widening relative to MOMP. U2OS cells expressing TFAM-mClover, Omi-mCherry and $JF_{646}$-MOM were treated with ABT-737 in combination with the MCL1 inhibitor S63845 in the presence of qVD-OPh then subject to live-cell imaging (Fig 3B, Video EV4). Analysis of individual mitochondria demonstrated, in a sequential manner, that mitochondrial release of Omi-mCherry (denoting MOMP) occurs prior to the visual appearance and widening of MOM pores (Fig 3C and D), finally followed by TFAM-mClover release. Importantly, not all mitochondria following MOMP released TFAM and, when it did occur, the time between MOMP and release of TFAM was highly variable, ranging from 10 to 100 min (Fig EV2C and D).

## Mitochondrial inner membrane permeabilisation allows mtDNA release into the cytosol

Thus far, our data demonstrate that mtDNA/TFAM complexes relocalise beyond the mitochondrial outer membrane; however, the integrity of the inner mitochondrial membrane (IMM) or localisation of mtDNA relative to the IMM was not determined. To investigate this, U2OS cells were treated with ABT-737/ActD in the presence of qVD-OPh, immunostained for apoptosis-inducing factor (AIF) and DNA (to visualise the IMM and mtDNA, respectively) and analysed by Airyscan imaging (Fig 4A, Videos EV5 and EV6). In healthy cells, the majority of mtDNA signal was within AIF immunostained structures, in line with AIF being inner membrane localised and mtDNA residing within the matrix (Fig 4A, Video EV5). Importantly, following MOMP (ABT-737/ActD/qVD-OPh treatment), mtDNA was found to localise in the cytoplasm, outside the inner membrane (AIF immunostained structures; Fig 4A, Video EV6). This suggests that mtDNA can pass beyond the inner membrane during caspase-independent cell death (CICD). To investigate this further, we simultaneously imaged the MOM, IMM and mtDNA during cell death. U2OS cells stably expressing $JF_{646}$-MOM were treated with ABT-737/ActD/qVD-OPh and then immunostained with anti-AIF and anti-DNA antibodies to detect the MOM, IMM and mtDNA, respectively. Cells were analysed by Airyscan imaging and 3D-structured illumination microscopy (3D-SIM; Figs 4B and EV3A). 3D-SIM analysis demonstrated re-localisation of both AIF and mtDNA beyond the mitochondrial outer membrane specifically following MOMP (Fig EV3A). Consistent with this, Airyscan analysis also demonstrated AIF and mtDNA re-localisation beyond the mitochondrial outer membrane specifically following MOMP (Fig 4B). 3D-Imaris-based analysis revealed mtDNA and inner membrane protrusion through the mitochondrial outer membrane following MOMP (Figs 4C and EV3B), in line with earlier live-cell analysis of TFAM-mClover, showing its extrusion

---

**Figure 2.  MOMP-induced mtDNA release initiates a cGAS-STING-dependent type I interferon response.**

A   SVEC cells treated with 10 μM ABT-737 and 10 μM S63845 ± 20 μM qVD-OPh. Cell viability was analysed using an IncuCyte live-cell imager and SYTOX Green exclusion. Data are expressed as mean ± SEM, representative of two independent experiments.

B   Airyscan images of SVEC cells treated with 10 μM ABT-737, 10 μM S63845 and 20 μM qVD-OPh for 3 h, immunostained with anti-TOM20 and anti-DNA antibodies. Scale bar = 10 μm. Representative images from three independent experiments.

C   *Ifnb1*, *Irf7* and *Oasl1* mRNA expression in SVEC cells treated with 10 μM ABT-737, 10 μM S63845 and 20 μM qVD-OPh for 3 h. Data are representative of three independent experiments.

D   *Ifnb1* mRNA expression in SVEC cells treated with 10 μM ABT-737, 10 μM S63845 ± 20 μM qVD-OPh for 2 h. Data are representative of two independent experiments.

E   STING expression in CRISPR-Cas9-mediated STING-deleted SVEC cells using three independent sgRNA sequences.

F   *Ifnb1* mRNA expression in STING CRISPR-Cas9-deleted SVEC cells treated with 10 μM ABT-737, 10 μM S63845 and 20 μM qVD-OPh for 3 h. Data are representative of two independent experiments.

G   BAX and BAK expression in SVEC cells harbouring CRISPR-Cas9-mediated deletion of BAX, BAK or BAX/BAK.

H   *Ifnb1* mRNA expression in BAX, BAK or BAX/BAK CRISPR-Cas9-deleted SVEC cells treated with 10 μM ABT-737, 10 μM S63845 and 20 μM qVD-OPh for 3 h. Data are representative of two independent experiments.

Source data are available online for this figure.

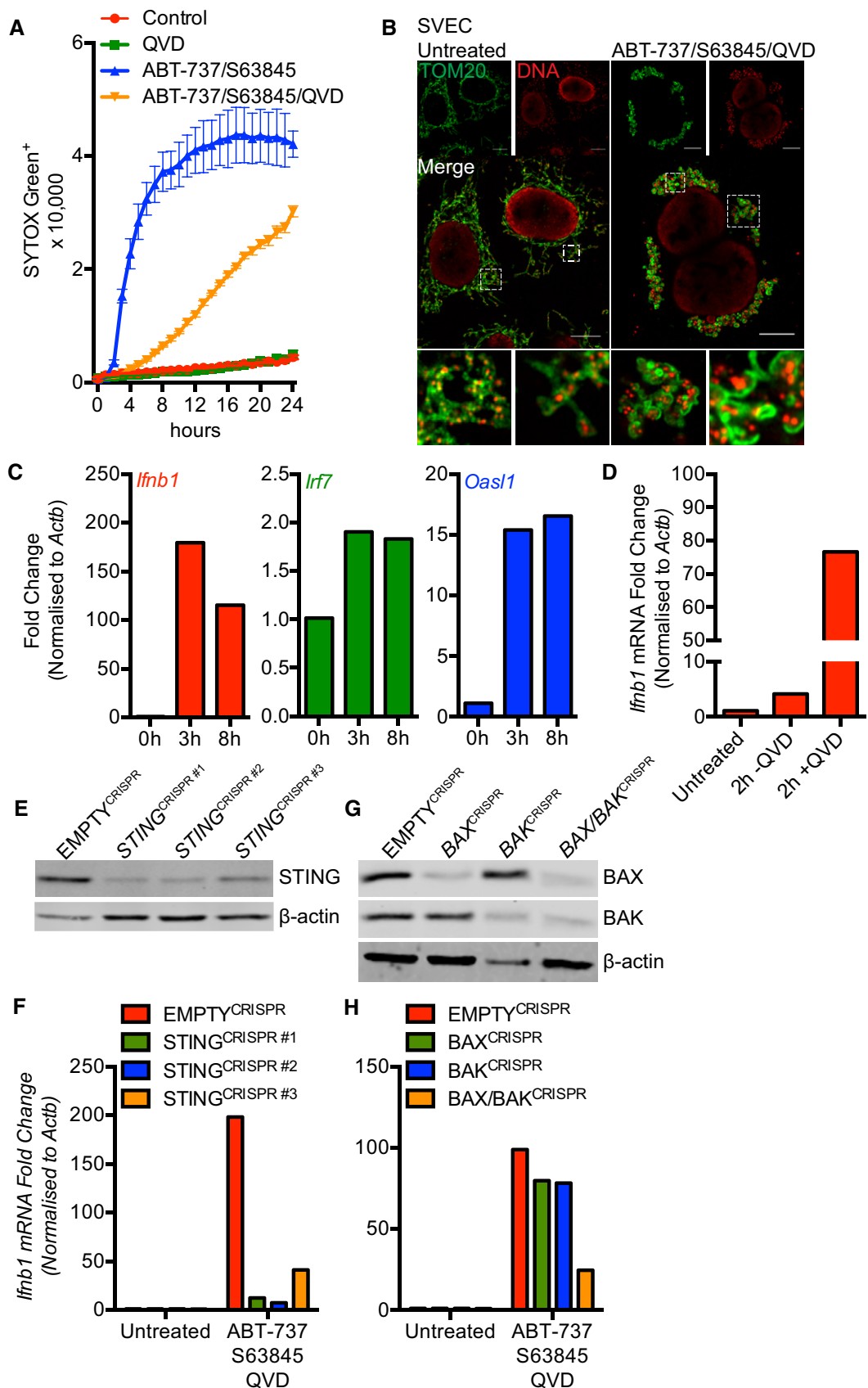

**Figure 2.**

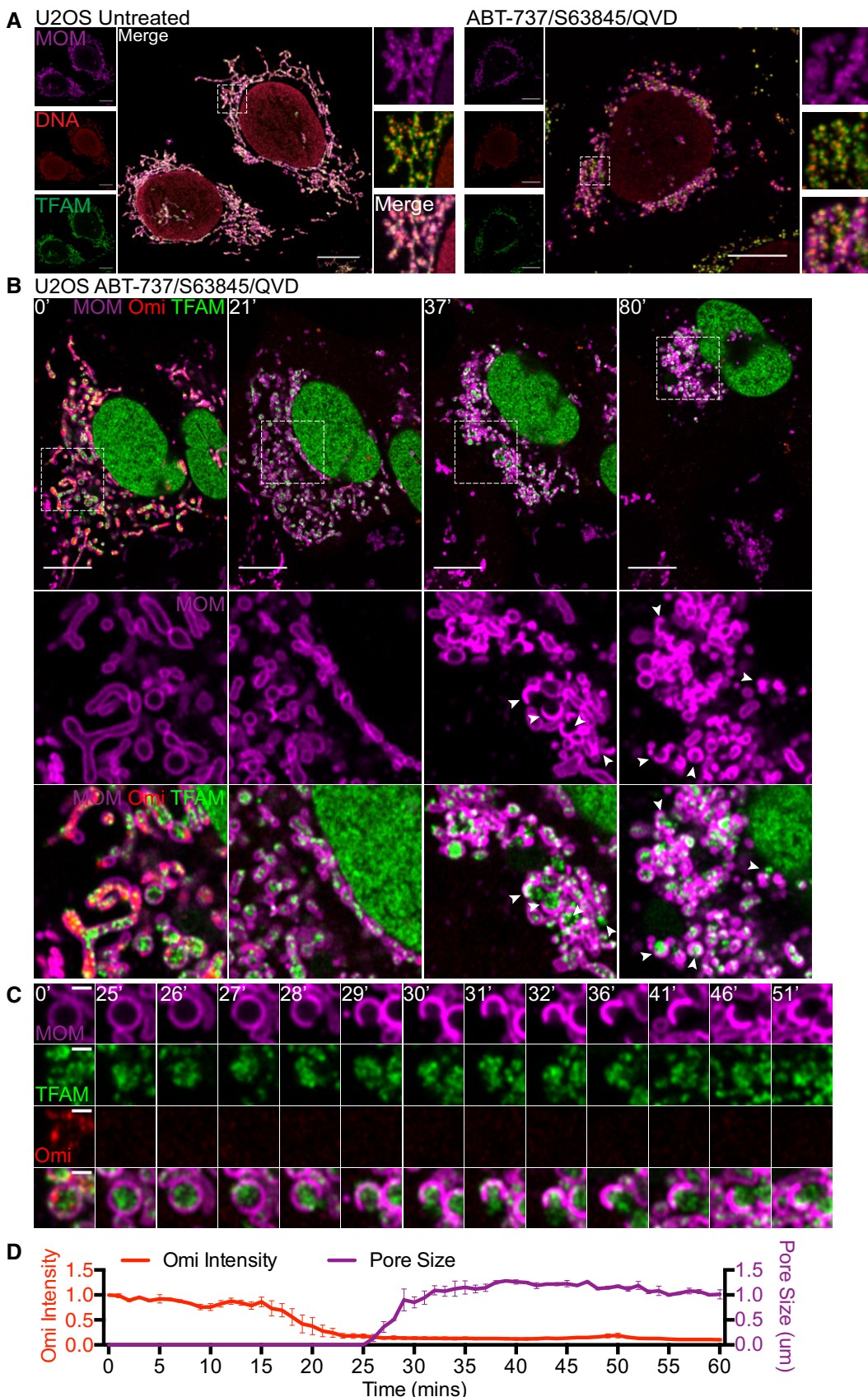

**Figure 3.**

**Figure 3.  mtDNA extrusion occurs through expanding outer membrane pores.**

A    Airyscan images of U2OS cells stably expressing JF$_{646}$-MOM (magenta) treated with 10 μM ABT-737, 2 μM S63845 and 20 μM qVD-OPh for 3 h, immunostained with anti-DNA and anti-TFAM antibodies. Scale bar = 10 μm. Representative images from two independent experiments.

B    Live-cell Airyscan imaging of U2OS cells stably expressing JF$_{646}$-MOM (magenta) and Omi-mCherry (red) and transiently expressing TFAM-mClover (green). Cells were treated with 10 μM ABT-737, 10 μM S63845 and 20 μM qVD-OPh at $t = 0$. Arrows denote mitochondria which show TFAM release. Scale bar = 10 μm. See Video EV4. Numbers indicate time in minutes.

C    Zoom of a single mitochondrion from Fig 3B and Video EV4. Scale bar = 1 μm. Numbers indicate time in minutes.

D    Loss of Omi intensity as assessed by standard deviation of the Omi signal across the cell and pore size plotted against time. Data are from three mitochondria (pore size) or three cells (Omi release) from two independent experiments and are expressed as mean ± SEM.

Source data are available online for this figure.

(Fig 3B, Videos EV2–EV4). In agreement with our earlier data, mtDNA also relocalised outside the mitochondrial outer membrane following MOMP, either associated or not with the inner membrane (Fig EV3B). Next, we aimed to visualise this by live-cell microscopy. To this end, we co-expressed JF$_{646}$-MOM, AIF (1-90)-mClover and TFAM-mScarlet in U2OS cells to visualise the MOM, IMM and matrix, respectively. To induce CICD, cells were treated with ABT-737/S63845 in the presence of qVD-OPh and analysed by live-cell microscopy (Fig 4D, Video EV7). This approach enabled specific visualisation of the MOM and IMM; TFAM localised within the IMM, consistent with its matrix localisation (Fig 4D, 0 min). Similar to previous results, over time large pores in the MOM appeared, preceding extrusion of IMM enclosed matrix (Fig 4D, 60 min). Finally, TFAM was found outside the IMM (Fig 4D, 120 min). Together, these data suggest that following MOMP, the mitochondrial inner membrane is extruded through the permeabilised outer membrane. Mitochondrial inner membrane permeabilisation, or MIMP, can then occur, allowing mtDNA egress into the cytosol.

**Mitochondrial mtDNA release is independent of mitochondrial dynamics and permeability transition**

Various studies have shown that upon MOMP the GTPase DRP-1 promotes mitochondrial network fragmentation and inner membrane remodelling (Desagher & Martinou, 2000; Frank *et al*, 2001; Estaquier & Arnoult, 2007). Potentially, inner membrane remodelling and/or mitochondrial fission at the point of MOMP might facilitate MIMP and mtDNA release. To investigate a role for DRP-1, we used conditional Drp1$^{fl/fl}$ MEFs, using adenovirus expressing Cre recombinase to effectively delete DRP-1 protein (Fig EV4A; Wakabayashi *et al*, 2009). Wild-type or DRP-1-deleted MEF were then treated with ABT-737/ActD to engage

mitochondrial apoptosis. Cells were imaged using IncuCyte live-cell imaging and SYTOX Green uptake. ABT-737/ActD treatment effectively induced apoptosis to a similar extent irrespective of DRP-1 expression (Fig EV4B). In the presence of caspase inhibitor, samples were immunostained for TOM20 and mtDNA, and identified as having undergone MOMP by loss of mitochondrial cytochrome *c* staining. 3D images were generated from *Z*-stacks of high-resolution images and quantified for mtDNA release (Fig 5A and B). In control samples, DRP-1-deleted cells displayed hyperfused mitochondria, as expected (Fig 5A). Importantly, the extent of mtDNA release in cells that had undergone MOMP failed to reveal a difference between wild-type and DRP-1-deficient cells (Fig 5A and B). To investigate a role for mitochondrial fission using an alternative method, we investigated whether mitochondria that have already undergone fission could release mtDNA upon MOMP. To this end, U2OS MCL-1$^{CRISPR}$ cells were pre-incubated with carbonyl cyanide *m*-chlorophenyl hydrazine (CCCP) to induce extensive mitochondrial fission and then treated to undergo MOMP by addition of ABT-737 in the presence of qVD-OPh. Cells were immunostained for TOM20 and mtDNA, then analysed by Airyscan microscopy. As expected, CCCP treatment induced extensive mitochondrial fragmentation (Fig 5C). Importantly, induced mitochondrial fission had no effect on MOMP-induced mtDNA release, consistent with the lack of requirement for DRP-1. To further define MIMP, we adapted an assay to measure mitochondrial release of matrix calcein relative to MOMP by live-cell imaging (Bonora *et al*, 2016). We reasoned that should the inner membrane become permeable, $Co^{2+}$ would be free to enter the matrix and quench matrix-located calcein, enabling of detection of increased permeability in real time. To simultaneously visualise MOMP, we expressed Omi-mCherry which resides in the mitochondrial inner membrane space (Tait *et al*, 2010). U2OS cells loaded with calcein-AM and $Co^{2+}$ were treated with ABT-737/

**Figure 4.  Mitochondrial inner membrane permeabilisation allows mtDNA release into the cytosol.**

A    Maximum intensity projection of Airyscan *z*-stack data of U2OS cells treated with 10 μM ABT-737, 1 μM ActD and 20 μM qVD-OPh for 3 h. Cells were immunostained with anti-AIF (IMM) and anti-DNA antibodies. *Z*-stack data were 3D-rendered in Imaris and mtDNA nucleoid inside and outside the AIF signal were visualised. Scale bar = 5 μm. See Videos EV5 and EV6. Representative images from three independent experiments.

B    Airyscan images of U2OS cells stably expressing JF$_{646}$-MOM (magenta) treated with 10 μM ABT-737, 1 μM ActD and 20 μM qVD-OPh for 3 h and immunostained with anti-DNA (red) and anti-AIF (IMM, green) antibodies. Scale bar = 10 μm. Representative images from three independent experiments.

C    Imaris 3D-renderings of MOM (magenta), AIF (green) and DNA (red) from U2OS cells treated as in (B).

D    U2OS cells stably expressing JF$_{646}$-MOM (magenta) and transiently expressing TFAM-mScarlet (red) and AIF (1-90)-mClover (green) live-cell imaged by Airyscan. Cells were treated with 10 μM ABT-737, 2 μM S63845 and 20 μM qVD-OPh at $t = 0$. Scale bar = 10 and 2 μm for zooms. See Video EV7. Representative images from two independent experiments. Numbers indicate time in minutes. Arrowheads indicate instances of released TFAM.

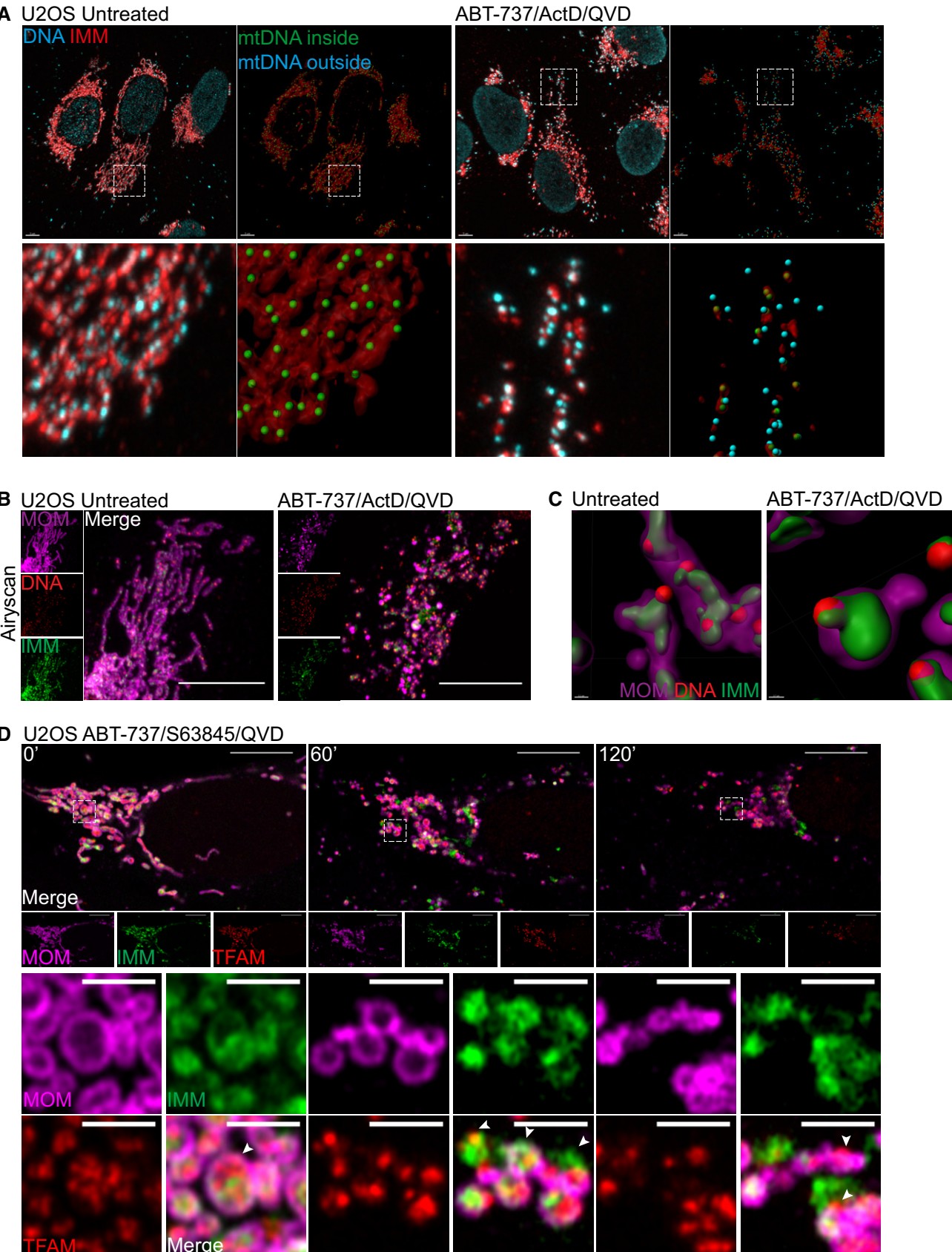

**Figure 4.**

**Figure 5.  mtDNA release is independent of fission and the mitochondrial permeability transition pore.**

A   Maximum intensity projection of *z*-stack Airyscan images of Drp1*fl/fl* MEFs with induced Drp1 deletion by AdCre lentiviral particles. Cells were treated with 10 μM ABT-737, 1 μM ActD and 20 μM qVD-OPh for 3 h and immunostained with anti-TOM20 (MOM) and anti-DNA antibodies. Zooms show Imaris 3D reconstructions of surface (MOM, TOM20) and spots (DNA) to show the extent of mtDNA release. Scale bar = 10 μm. Representative images from two independent experiments.

B   Quantification of mtDNA release per cell in *Wt* or *Drp1*-deleted cells. Data are from two independent experiments and are expressed as mean ± SD. Data were analysed by Student's *t*-test.

C   U2OS MCL1$^{CRISPR}$ cells were pre-treated with 10 μM CCCP and 20 μM qVD-OPh for 30 min to induce mitochondrial fragmentation after which treatment was changed to 10 μM ABT-737 and 20 μM qVD-OPh. After 3 h, cells were fixed and immunostained with anti-TOM20 and anti-DNA antibodies. Zooms show highlighted areas. Scale bar = 10 μm. Representative images from three independent experiments.

D   Images from time-lapse live-cell imaging of U2OS cells loaded with calcein-AM and CoCl$_2$ and treated with 10 μM ABT-737/S63845/qVD-OPh at *t* = 0. Scale bar = 10 μm. See Video EV8. Representative images from three independent experiments. Numbers indicate time in minutes.

E   Quantification of calcein release from mitochondria relative to Omi release. Data are from three independent experiments.

F   Maximum intensity projections of *z*-stack Airyscan images of Wt and *CypD$^{−/−}$* MEFs. Cells were treated with 10 μM ABT-737, 1 μM ActD and 20 μM qVD-OPh for 3 h and immunostained for TOM20 (MOM) and DNA. Scale bar = 10 μm. Representative images from two independent experiments.

G   Quantification of mtDNA release per cell in *Wt* or *CypD*-deleted cells. Data are from two independent experiments and are expressed as mean ± SD. Data were analysed by Student's *t*-test.

Source data are available online for this figure.

S63845 in the presence of qVD-OPh and imaged by confocal micro-scopy (Fig 5D and Video EV8). In healthy cells, mitochondrial localised calcein persisted, consistent with the mitochondrial IMM being impermeable (Fig EV4D). However, following MOMP (de-fined by mitochondrial release of Omi-mCherry) the matrix calcein signal was rapidly lost within minutes (mean time of 10 min) following MOMP (Fig 5D and E, and Video EV8). Loss of calcein signal was not blocked by co-treatment with the cyclophilin D inhibitor cyclosporin A (CsA; Fig EV4E and Video EV9). This demonstrates, at early time points following MOMP, the mitochon-drial inner membrane increases permeability to small ions. Prolonged opening of the mitochondrial permeability transition pore (MPTP) causes mitochondrial swelling leading to mitochon-dria rupture (Izzo *et al*, 2016). We therefore investigated whether MPTP contributed to mtDNA release. Wild type and MEFs deleted for cyclophilin D (Fig EV4C; a critical component of the MPTP; Baines *et al*, 2005; Nakagawa *et al*, 2005) were treated with ABT-737/ActD/qVD-OPh to engage MOMP and then immunostained for DNA and TOM20, to visualise mtDNA and the mitochondrial outer membrane, respectively. Importantly, a similar extent of mtDNA release was observed between WT and CypD$^{−/−}$ cells, ruling out a critical role for MPTP in mtDNA release (Fig 5F and G). Collec-tively, these data show that MIMP, enabling the release of mtDNA, is independent of both mitochondria dynamics and MPTP.

**Mitochondrial inner membrane is extruded through expanding BAX pores**

We next sought to understand how the inner membrane breaches the outer membrane, permitting mtDNA release. It is well estab-lished that during apoptosis BAX and BAK form oligomers on the mitochondrial outer membrane leading to MOMP (Cosentino & Garcia-Saez, 2017). We therefore investigated the relationship between BAX activation, MOMP and MIMP through live-cell imag-ing. To this end, mCherry-BAX was expressed in BAX/BAK-deleted U2OS cells together with JF$_{646}$-MOM and TFAM-mClover. Cells were treated with ABT-737/S63845 in the presence of qVD-OPh to induce MOMP then analysed by live-cell imaging (Figs 6A and EV5A, Video EV10). Consistent with our earlier data, following treatment, pores in the MOM occurred that gradually increased in size (Figs 6A and EV5A, Video EV10). Combined analysis of the MOM, TFAM and

BAX showed that TFAM is commonly extruded from large MOM openings that are decorated by BAX puncta. This suggests that growing BAX-mediated pores in the mitochondrial outer membrane enables extrusion of the mitochondrial inner membrane. To investi-gate this further, we treated U2OS MCL1$^{CRISPR}$ cells (which die in a rapid and synchronous manner in response to ABT-737) expressing JF$_{646}$-MOM and TFAM-mClover with ABT-737 in the presence of qVD-OPh and immunostained with antibody recognising active BAX (6A7; Hsu & Youle, 1997). In line with previous findings, we found that during MOMP, the MOM becomes decorated in activated BAX punctate structures (Figs 6B and EV5B). 3D images were generated from *Z*-stacks of super-resolution images (Figs 6C and EV5C). These images frequently showed translocation of TFAM-mClover through the mitochondrial outer membrane at sites demarked by active BAX staining. We next investigated the inner membrane relative to BAX localisation and outer membrane opening. U2OS cells expressing JF$_{646}$-MOM were treated with ABT-737/S63845 in the presence of qVD-OPh and immunostained for active BAX (6A7) and AIF to detect the inner membrane (Figs 6D and EV5D). Specifically follow-ing treatment, the inner membrane was found to be extruded via active BAX decorated pores. This was supported by 3D-Imaris-based analysis (Figs 6E and EV5E). These data support a model, whereby following MOMP the mitochondrial inner membrane can be extruded through BAX-mediated outer membrane pores. Once in the cytoplasm, the inner membrane can permeabilise, allowing mtDNA to activate cGAS-STING signalling.

# Discussion

Following apoptotic mitochondrial permeabilisation, mtDNA acti-vates cGAS-STING-dependent interferon synthesis under caspase-inhibited conditions (Rongvaux *et al*, 2014; White *et al*, 2014; Giampazolias *et al*, 2017). Here, we investigated how matrix loca-lised mtDNA could activate cytosolic cGAS-STING signalling, given that their separate subcellular localisation. We find that under caspase-inhibited conditions, BAX/BAK-mediated MOMP allows extrusion of the mitochondrial inner membrane into the cytoplasm. Unexpectedly, the extruded inner membrane eventually perme-abilises leading to cytosolic release of mtDNA where it can activate cGAS-STING signalling. By doing so, mitochondrial inner membrane

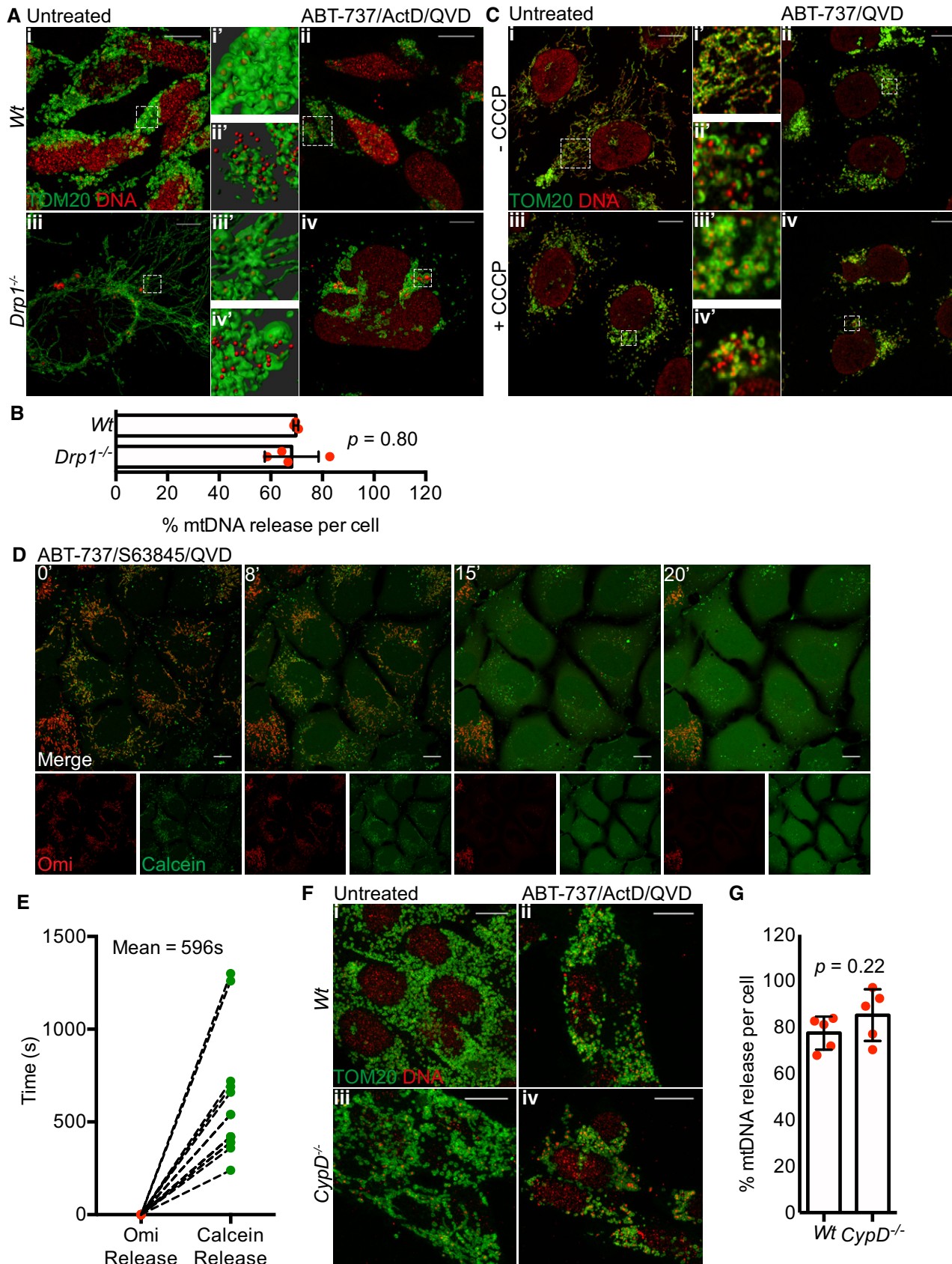

**Figure 5.**

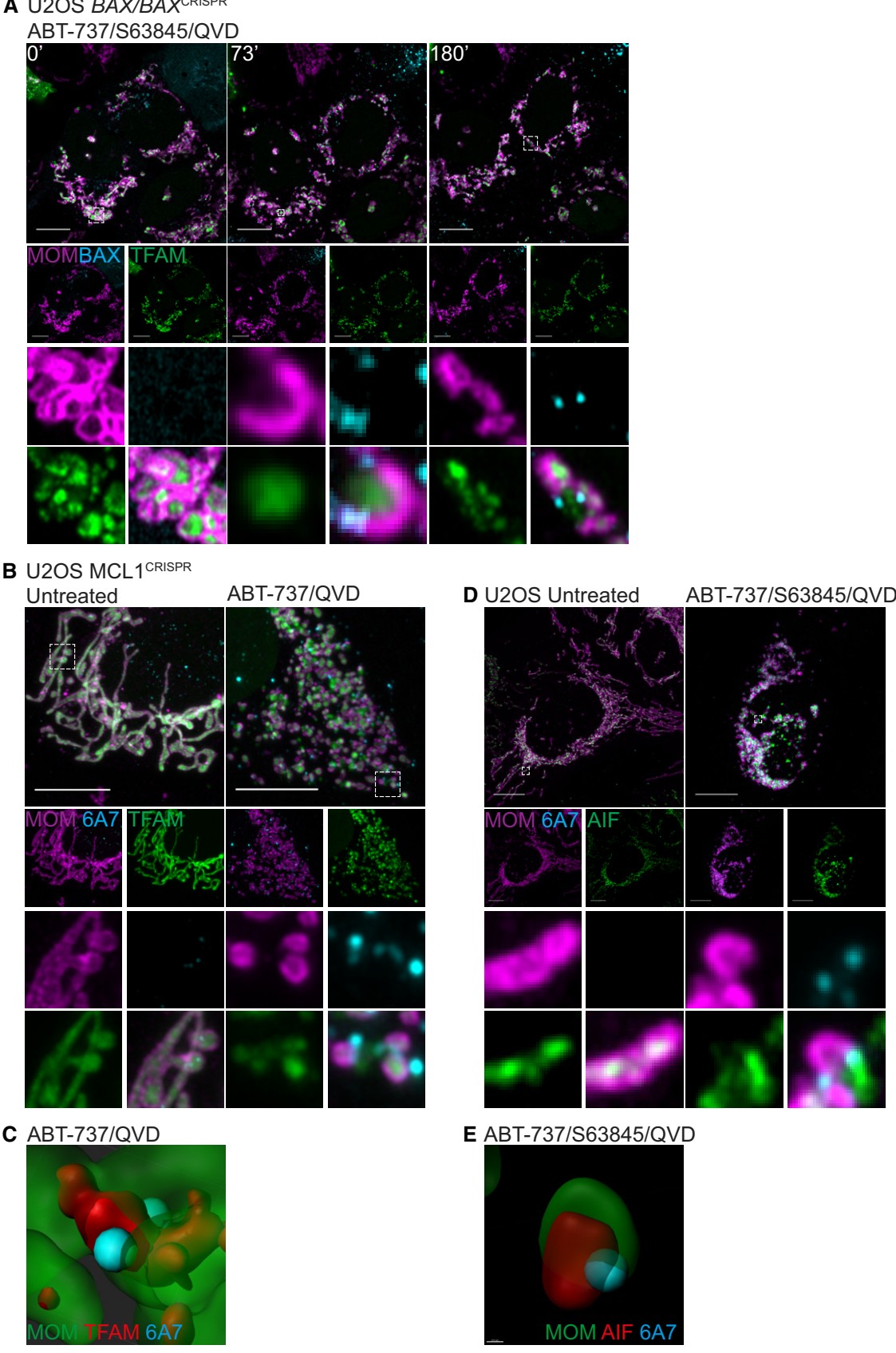

**Figure 6.**

**Figure 6.  Mitochondrial inner membrane and mtDNA are extruded through BAX pores on the mitochondrial outer membrane.**

A  Live-cell Airyscan images of U2OS *BAX/BAK* CRISPR-Cas9-deleted cells stably expressing JF$_{646}$-MOM (magenta) and mCherry-BAX (cyan) and transiently expressing TFAM-mClover (green). Cells were treated with 10 μM ABT-737, 2 μM S63845 and 20 μM qVD-OPh at $t$ = 0. Scale bar = 10 μm. See Video EV10. Numbers indicate time in minutes.

B  Airyscan images of U2OS MCL1$^{CRISPR}$ cells stably expressing JF$_{646}$-MOM (magenta) and transiently expressing TFAM-mClover (green), treated with 10 μM ABT-737 and 20 μM qVD-OPh for 3 h, immunostained with anti-active BAX (6A7, cyan). Images are maximum intensity projections of *z*-stack data. Scale bar = 10 μm. Representative images from three independent experiments.

C  Imaris 3D-rendering of U2OS cells as in (B) showing MOM (green), TFAM (red) and active BAX (cyan).

D  U2OS cells stably expressing JF$_{646}$-MOM (magenta) treated with 10 μM ABT-737, 2 μM S63845 and 20 μM qVD-OPh for 3 h. Cells were immunostained with antibodies for active BAX (6A7, cyan) and AIF (IMM, green). Images are maximum intensity projections of *z*-stack data. Scale bar = 10 μm. Representative images from three independent experiments.

E  Imaris 3D-rendering of U2OS cells as in (D) showing MOM (green), AIF (red) and active BAX (cyan).

permeabilisation or MIMP supports the immunogenic effects of caspase-independent cell death.

Surprisingly, our data show that mitochondrial inner membrane can permeabilise subsequent to BAX/BAK-mediated MOMP. We found that over time, following MOMP (defined by release of Omi-mCherry), visible and widening pores appeared in the MOM through which the inner membrane is extruded (Figs 3C and D, and 4). Interestingly, active BAX was found to localise at the edge of the widening MOM pores through which the inner membrane extrudes (Fig 6). This is reminiscent of recent studies showing that during apoptosis, BAX-rings border visible pores in the mitochondrial outer membrane (Grosse *et al*, 2016; Salvador-Gallego *et al*, 2016). Over time, we observed these MOM pores gradually widen enabling inner membrane extrusion. Coupling our data to recent findings demonstrating variation in BAX-ring size (Salvador-Gallego *et al*, 2016) as well as wide variation in BAX-mediated pore size (Dewson *et al*, 2012; Czabotar *et al*, 2013) strongly suggests that BAX-mediated pores are dynamic and grow over time. This is consistent with growing BAX pores previously detected during BAX-mediated liposomal permeabilisation (Bleicken *et al*, 2013).

How the inner membrane eventually permeabilises remains unclear. Our data argue against any role for mitochondrial permeability transition or mitochondrial dynamics in the process. Inner membrane permeabilisation occurs considerably later (1 h) than MOMP, with less penetrance and in apparent stochastic manner (Fig 5E). As such, whether MIMP occurs in a regulated manner is unclear. Recently, McArthur, Kile and colleagues also described inner membrane permeabilisation during apoptosis and CICD enabling release of mtDNA (McArthur *et al*, 2018). The extent of MIMP they observed was significantly less than us, potentially reflecting cell-type differences in the propensity of mitochondria to undergo MIMP. Interestingly, we detected increased permeability of the mitochondrial inner membrane to small ions, shortly following the onset of MOMP. This increased permeability may explain the transient drop in inner membrane potential immediately following MOMP previously reported by others (Waterhouse *et al*, 2001). What drives this increased permeability is unknown but we reason it must be transient, since many studies have shown mitochondrial inner membrane potential can be maintained post-MOMP under caspase-inhibited conditions (Bossy-Wetzel *et al*, 1998; von Ahsen *et al*, 2000; Waterhouse *et al*, 2001; Huber *et al*, 2011). While inner membrane extrusion and MIMP can be temporally separated from mtDNA release, it is possible that the increased permeability of the inner membrane we observe following MOMP may precipitate these later events.

Following MOMP, our ability to detect inner membrane extrusion and MIMP during cell death was dependent upon inhibition of caspase function. This contrasts with findings from McArthur, Kile and colleagues, whom could detect mtDNA release during (caspase-dependent) apoptosis (McArthur *et al*, 2018). Notably, induction of mitochondrial apoptosis under caspase-proficient conditions also led to an increase in IFN-β mRNA, albeit not to the same extent as under conditions of CICD (Fig 2D). We therefore think it likely that our inability to visualise MIMP during apoptosis relates to the kinetics of apoptotic cell death being much more rapid than MIMP, such that rounding and detachment of apoptotic cells made them impossible to image over a longer period (Video EV1). Even if MIMP does occur during apoptosis, its functional effects may be limited due to rapid loss of cell viability coupled to the strong inhibitory effects of caspase activity on protein translation (Clemens *et al*, 2000). Indeed, we and others have shown that MOMP has pro-inflammatory effects only under caspase-inhibited conditions (Rongvaux *et al*, 2014; White *et al*, 2014; Giampazolias *et al*, 2017). Nevertheless, physiological engagement of MIMP, leading to mtDNA release and interferon synthesis, might be expected in cell types deficient or defective in caspase activity such as cardiomyocytes and sympathetic neurons following MOMP (Potts *et al*, 2005; Wright *et al*, 2007; Potts *et al*, 2005; Wright *et al*, 2007). Another possibility relates to our recent finding that sub-lethal stress can engage MOMP in a limited cohort of mitochondria without cell death (Ichim *et al*, 2015). Whether MIMP and mtDNA release occurs in these permeabilised mitochondria leading to cGAS-STING activation is currently unclear.

Intense interest currently surrounds the pharmaceutical activation of cGAS-STING signalling to improve cancer immunotherapy (Ng *et al*, 2018). Our own work has shown that engaging MOMP under caspase-inhibited conditions—caspase-independent cell death (CICD)—can potently activate anti-tumour immunity pathway (Giampazolias *et al*, 2017). The inflammatory effects of CICD are dependent on recognition of mtDNA by the cytosolic cGAS-STING sensing pathway (Rongvaux *et al*, 2014; White *et al*, 2014; Giampazolias *et al*, 2017). Underlying this effect, here we have shown that following MOMP, the inner membrane is extruded via BAX-mediated pores prior to permeabilisation and mtDNA release. mtDNA-dependent activation of cGAS-STING has recently been implicated in variety of pathophysiological processes including infectious and inflammatory diseases (West & Shadel, 2017). As such, further understanding the mechanism(s) of mtDNA mitochondrial release in different contexts may provide new avenues for therapeutic exploitation.

# Materials and Methods

## Cell lines, plasmids, reagents and antibodies

U2OS, MEF, SVEC and 293FT were cultured in DMEM high-glucose medium supplemented with 10% FCS, 2 mM glutamine, 1 mM sodium pyruvate, 50 μM β-mercaptoethanol, penicillin (10,000 units/ml) and streptomycin (10,000 units/ml). Reagents used were as followed: ABT-737, Q-VD-OPh (APEX Bio), S63845 (Active Biochem) carbonyl cyanide 3-chlorophenylhydrazone (CCCP), cyclosporin A (Sigma), actinomycin D (Merck). Cell lines were routinely tested for the presence of mycoplasma. No cell lines were authenticated, although SVEC cells were newly obtained from ATCC (LGC Standards). To delete Drp1 from Drp1$^{fl/fl}$ MEFs, $2 \times 10^6$ cells were seeded and infected with 200 MOI high titre Ad5CMVCre (Viral Vector Core, University of Iowa) for 8 h, after which time the media was removed and replaced. Cells were seeded the following day for experiments. For CRISPR/Cas9-based genome editing, the following sequences were cloned into LentiCRISPRv2-puro (Addgene #52961) or LentiCRISPRv2-blasti (Lopez *et al*, 2016):

Human BAX: 5′-AGTAGAAAAGGGCGACAACC-3′
Human BAK: 5′-GCCATGCTGGTAGACGTGTA-3′
Human MCL1: 5′-GGGTAGTGACCCGTCCGTAC-3′
Mouse BAX: 5′-CAACTTCAACTGGGGCCGCG-3′
Mouse BAK: 5′-GCGCTACGACACAGAGTTCC-3′
Mouse STING #1: 5′-CAGTAGTCCAAGTTCGTGCG-3′
Mouse STING #2: 5′-AGCGGTGACCTCTGGGCCGT-3′
Mouse STING #3: 5′-GTTAAATGTTGCCCACGGGC-3′

For lentiviral and retroviral transduction, the following plasmids were used: pLJM2 SNAP-Omp25 was obtained from Addgene (#69599). mCherry-BAX was cloned into the pMx vector with blasticidin resistance using Gibson assembly.

Transient expression vectors were generated as follows: pcDNA3 TFAM-mClover3 and pcDNA3 TFAM-mScarlet were cloned using Gibson assembly (New England Biolabs) using TFAM-GFP (kind gift from Mikhail Alexeyev, University of South Alabama (Pastukh *et al*, 2007), pKanCMV-mclover3-18aa-actin (Addgene #74259) and pmScarlet-i_H2A_C1 (Addgene #85053) as templates. pcDNA3 AIF (1-90)-mClover3 was cloned using Gibson assembly using pcDNA AIF (1-90)-mCherry (Ichim *et al*, 2015) and pKanCMV-mclover3-18aa-actin as templates.

Primary antibodies for Western blotting were as follows: rabbit anti-BAX, rabbit anti-BAK, rabbit anti-DRP1, rabbit anti-STING (Cell Signaling), mouse MitoProfile Membrane Integrity Cocktail (for cyclophilin D; Abcam) and β-actin (Sigma). Primary antibodies for immunofluorescence were as follows: rabbit anti-TOM20 (Santa Cruz), mouse anti-DNA (Progen), rabbit anti-AIF, rabbit anti-TFAM (Cell Signaling), mouse anti-cytochrome *c* (BD Transduction Laboratories), mouse anti-active BAX (6A7, Santa Cruz).

## Stable cell line generation

For retroviral transduction, 293FT cells were transfected with 5 μg of selected plasmid in combination with 1.2 μg gag/pol (Addgene #14887) and 2.4 μg UVSVG (Addgene #12260) using Lipofectamine 2000 (Life Technologies). Viral supernatant was collected, filtered and used to infect cells 24 and 48 h post-transfection in the presence of 1 μg/ml Polybrene (Sigma-Aldrich). Cells were selected by addition of 1 μg/ml puromycin (Sigma) or 10 μg/ml blasticidin (InvivoGen) as appropriate. For cell lines stably overexpressing fluorescent proteins, cells were FACS sorted to isolate a high-expressing population.

For lentiviral transduction, 293FT cells were transfected with 5 μg of selected plasmid in combination with 1.86 μg psPAX2 (Addgene #8454) and 1 μg UVSVG (Addgene #12260) using Lipofectamine 2000. Viral supernatant was collected, filtered and used to infect cells 24 and 48 h post-transfection in the presence of 1 μg/ml Polybrene. Cells were selected by addition of 1 μg/ml puromycin. For cell lines stably overexpressing fluorescent proteins, cells were FACS sorted to isolate a high-expressing population.

## Clonogenic survival assay

U2OS cells were seeded at a density of 500 cells/well in 6-well plates and left to adhere overnight. The following morning, cells were treated as indicated and media replaced after 24 h. Cells were left to grow and form colonies for 10 days before fixation with ice-cold methanol and staining with crystal violet.

## qRT–PCR

RNA was isolated from cells using GeneJET RNA Purification Kit (Thermo Fisher Scientific) and included a DNase I treatment step (Roche) to remove genomic DNA contamination. cDNA was generated from RNA using the High-Capacity cDNA Reverse Transcription kit (Thermo Fisher Scientific). PCR was performed on a Bio-Rad C1000 Thermal Cycler using the following conditions: 3 min at 95°C, 40 cycles of 20 s at 95°C, 30 s at 57°C, 30 s at 72°C and a final 5 min at 72°C using the Brilliant III Ultra-Fast SYBR Green qPCR Master Mix (Agilent Technologies). mRNA quantification was analysed using the $2^{-\Delta\Delta CT}$ methods. Primer sequences used are as follows:

Mouse Ifnb1-F: AACTCCACCAGCAGACAGTG
Mouse Ifnb1-R: AGTGGAGAGCAGTTGAGGAC
Mouse Irf7-F: TTTGGAGACTGGCTATTGGGG
Mouse Irf7-R: CCTACGACCGAAATGCTTCCA
Mouse Oasl1-F: CATGCTCCCAAGCTTCTCTCT
Mouse Oasl1-R: CTGCCATGGCTCCTCCTTTTT
Mouse Actin-F: CACTGTCGAGTCGCGTCC
Mouse Actin-R: GTCATCCATGGCGAACTGGT

## Microscopy

Cells were fixed in 4% PFA/PBS for 10 min, followed by permeabilisation in 0.2% Triton X-100/PBS for 15 min, followed by blocking in 2% BSA/PBS for 1 h. Cells were incubated with primary antibodies overnight in a humidified chamber at 4°C followed by extensive washing before addition of secondary antibodies for 1 h at room temperature. Secondary antibodies used for detection were as follows: Alexa Fluor 488 goat anti-mouse (A11029), Alexa Fluor 488 goat anti-rabbit (A11034), Alexa Fluor 568 goat anti-mouse (A11004), Alexa Fluor 568 goat anti-rabbit (A11011), Alexa Fluor 647 goat anti-mouse (A21236), Alexa Fluor 647 goat anti-rabbit

(A21245); all purchased from Life Technologies. Slides were mounted in Vectashield antifade aqueous mounting medium (Vector Laboratories), with or without DAPI. For triple colour antibody-based immunofluorescence, cells were co-incubated with two antibodies (typically TOM20 and DNA), thoroughly washed and incubated with Alexa Fluor 488/568 secondary antibodies. Following extensive washing, cells were blocked with 5% mouse serum (Life Technologies) for 1 h at room temperature and further incubated with cytochrome *c* directly conjugated to Alexa Fluor 647 (clone 6H2.B4, Biolegend).

### Airyscan imaging

Super-resolution Airyscan images were acquired on a Zeiss LSM 880 with Airyscan microscope (Carl Zeiss). Samples were prepared on high precision cover glass (Zeiss, Germany). Data were collected using a $63 \times 1.4$ NA objective for the majority of experiments, except for calcein-AM experiments where the $40\times$ ($1.3$ NA) objective was used; 488, 561 and 640 nm laser lines were used, and refractive index-matched immersion oil (Zeiss) was used for all experiments. Where *z*-stacks were collected, the software-recommended optimal slice sizes were used. For both fixed-cell and live-cell experiments, colours were collected sequentially in order to minimise bleedthrough. Live-cell experiments were performed in an environmental chamber to allow maintenance of 37°C temperature and 5% $CO_2$. Airyscan processing was performed using the Airyscan processing function in the ZEN software. To maintain clarity and uniformity throughout the paper, some images have been pseudo-coloured.

### 3D-SIM imaging

3D structured illumination microscopy (3D-SIM) images were acquired on a Nikon N-SIM microscope (Nikon Instruments, UK). Samples were prepared on high precision cover glass (Zeiss, Germany). Data were collected using a Nikon Plan Apo TIRF $100 \times 1.49$ NA objective and an Andor DU-897X-5254 camera using 488, 561 and 640 nm laser lines. Refractive index-matched immersion oil (Nikon Instruments) was used for all experiments. *Z*-stacks were collected with a step size of 120 nm as required by the manufacturer software. For each focal plane, 15 images were acquired (five phases, three angles) and captured using the NiS-Elements software. SIM image processing, reconstruction and analysis were carried out using the N-SIM module of the NiS-Elements Advanced Research software. Images were checked for artefacts using the SIMcheck software (http://www.micron.ox.ac.uk/software/SIMCheck.php). Images were reconstructed using NiS-Elements software v4.6 (Nikon Instruments, Japan) from a *z*-stack comprising of no < 1 μm of optical sections. In all SIM image reconstructions, the Wiener and Apodization filter parameters were kept constant.

### 3D renderings, mtDNA release quantification and image analysis

Acquired *z*-stacks were imported into Imaris (Bitplane, Oxford Instruments, Switzerland). To segment inner mitochondrial membrane (IMM) and mtDNA nucleoids, a surface was creating using the IMM channel. Masks were applied to differentiate between mtDNA nucleoids inside and outside the surface (i.e. in or out of the IMM). From these masks, spots were created from the mtDNA channel.

For quantification of mtDNA outside the mitochondrial outer mitochondrial (MOM), a surface was created for the MOM channel. Spots were then generated from the mtDNA channel, and the transparency of the MOM surface altered to allow visualisation of mtDNA nucleoid spots outside or inside the MOM. Cells were visually scored to quantitate the degree of mtDNA release.

All images were analysed in ImageJ, including maximum intensity projections, before figure panels were assembled in Adobe Illustrator. Omi release was assessed by the standard deviation of the Omi signal across a cell in ImageJ. Mitochondria pore size was measured using the ruler tool in ImageJ.

### SNAP-tag labelling

Cell lines stably expressing pLJM2 SNAP-Omp25 were incubated with 15 nM $JF_{646}$ SNAP ligand (kind gift from Luke Lavis) for 30 min in complete medium. Throughout the paper, we refer to MOM labelled in this manner as $JF_{646}$-MOM. Cells were washed three times in medium and returned to the incubator for 15 min to allow any unbound dye to diffuse out. Media was replaced with FluoroBrite DMEM (Life Technologies) supplemented with 10% FCS, 2 mM glutamine and 1 mM sodium pyruvate and then imaged.

## Calcein-AM assay

Mitochondria inner membrane permeabilisation was assessed using a modified version of the calcein-AM/cobalt assay described by Bonora *et al* (2016). Briefly, U2OS cells stably expressing $JF_{646}$-MOM and Omi-mCherry were cells seeded onto coverglass dishes to 70% confluency. Cells were incubated with 1 μM of calcein-AM stock solution and 10 mM $CoCl_2$ in HBSS supplemented with 10 mM HEPES and 2 ml L-glutamine for 15 min at 37°C in a 5% $CO_2$ atmosphere. Cells were washed twice in HBSS, and then, 900 μl of supplemented HBSS and 10 mM $CoCl_2$ was added. Cells were left to acclimatise in the microscope environmental chamber for 30 min and imaged following addition of inducers of apoptosis, qVD-OPh $\pm$ cyclosporin A.

## Western blotting

Cells were lysed in NP-40 lysis buffer (1% NP-40, 1 mM EDTA, 150 mM NaCl, 50 mM Tris–Cl pH 7.4) supplemented with complete protease inhibitor (Roche). Protein concentration was determined by Bradford assay (Bio-Rad). Protein lysates were subjected to electrophoresis through 10 or 12% SDS–PAGE gels and transferred onto nitrocellulose membrane. Membranes were blocked in 5% milk/PBS-Tween for 1 h at room temperature and incubated with primary antibody over night at 4°C in blocking buffer. Following washing, membranes were incubated with Li-Cor secondary antibodies (IRDye 680RD donkey anti-mouse or IRDye 800CW donkey anti-rabbit) for 1 h at room temperature and protein levels detected using a Li-Cor Odyssey CLx (Li-Cor) and acquired using Image-Studio (Li-Cor). Resulting images were minimally processed for clarity and arranged using Adobe Illustrator.

## Live-cell viability assays

Cell viability was assayed using an IncuCyte FLR imaging system (Sartorius). Briefly, cells were seeded overnight and drugged in the

presence of 30 nM SYTOX Green (Life Technologies), a non-cell-permeable nuclear stain. Where the same cell line was compared, SYTOX Green-positivity was compared directly; where different cell lines (e.g. WT and KO MEFs) were compared, data were normalised to starting confluency. All quantification was performed using the IncuCyte software.

**Expanded View** for this article is available online.

## Acknowledgements

Funding for this work was from the BBSRC (grant BB/K008374/1) Cancer Research UK Programme Foundation Award (C40872/A20145; SWGT) and NIH grant GM123266 (HS). We thank Luke Lavis (HHMI/Janelia Research Campus), Mikhail Alexyev (University of South Alabama), David Sabatini (MIT) for reagents. We also thank Margaret O'Prey, David Strachan and Tom Gilbey (Beatson Institute) for excellent technical assistance, Catherine Winchester (Beatson Institute) and members of the Tait laboratory for critical reading of the manuscript. JC is funded by Centre for Ageing and Vitality supported by the BBSRC, EPSRC, ESRC, and MRC as part of the cross-council Lifelong Health and Wellbeing Initiative and JFP funded by BBSRC (grant BB/K017314/1).

## Author contributions

JSR and SWGT conceived the study and designed the workplan. Experimental work: JSR, CC, JC, MP, JFP, APW. Development and contribution of reagents: GQ, JL, JO'P, HS, KMR. Data analysis: JSR, JC, APW, SWGT Intellectual input: JSR, GQ, LMC, JFP, AO, SWGT Manuscript writing: JSR and SWGT.

## Conflict of interest

The authors declare that they have no conflict of interest.

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
