## [Review Process File · The EMBO Journal]

Mitochondrial Inner Membrane Permeabilisation Enables mtDNA Release During Apoptosis

Joel S Riley, Giovanni Quarato, Catherine Cloix, Jonathan Lopez, Jim O'Prey, Matthew Pearson, James Chapman, Hiromi Sesaki, Leo M Carlin, João F Passos, Ann P Wheeler, Andrew Oberst, Kevin M Ryan and Stephen WG Tait.

Review timeline:	Submission date:	19 February 2018
	Editorial Decision:	6 March 2018
	Revision received:	16 May 2018
	Editorial Decision:	6 June 2018
	Revision received:	11 June 2018
	Accepted:	18 June 2018

Editor: Elisabetta Argenzio

Transaction Report:

1st Editorial Decision

6 March 2018

Thank you for submitting your manuscript on the role of Bax/Bak in mitochondrial inner membrane permeabilization and mtDNA release during cell death. We have now received two referee reports on your manuscript, which are included below for your information.

As you can see, the referees consider the findings novel and of broad interest. However, they also raise some critical issues that need to be addressed before they can support publication here. In particular, they find that more evidence should be provided on how mtDNA activates the cGAS/STING pathway in the cytosol. In addition, they find that further quantitative and kinetic analysis is needed to formally prove the timing of MOM and MIM permeabilization and subsequent mtDNA release. Given the overall interest of your study, I would thus like to invite you to revise the manuscript in response to the referee reports. Please note that it is The EMBO Journal policy to allow only a single major round of revision and that it is therefore important to resolve the main concerns at this stage.

Thank you again for the opportunity to consider this work for publication, and please feel free to contact me with any questions about submission of the revised manuscript to The EMBO Journal. I look forward to your revision.

REFeree REPORTS

Referee #1:

I think the paper definitely deserves publication as the observations are very impressive, of broad interest, and they strengthen the findings from McArthur et al Science 2018. They come to similar

conclusion than the Science paper by using comparable, though different, methods. However there are a couple of points that should be considered:

1. From their data they find out that the IMM gets permeabilized allowing the release of DNA nucleoids. How do they explain that the DNA doesn't diffuse at a certain point in the cytosol away from the mitochondria? In the Science paper they talk about IMM herniation with few permeabilization events, still enough to trigger the cGas-STING signaling pathway. How do they reconcile their findings with the ones from McArthur et al? As they do not perform live cell imaging of the IMM, could their observation be due to methodological artifacts and/or to resolution limitation? The authors should in the best case perform live cell imaging of stained IMM, or alternatively, provide a valid explanation in the discussion section to the above mentioned issue. Also very important, the authors do not provide any evidence how mtDNA comes into contact with cGAS/STING, their conclusion is speculative and should be moderated.

2. The title of the article 'Activated Bax/Bak enable mitochondrial inner membrane permeabilisation and mtDNA release' is misleading, since there are no results about Bak in this manuscript. The title should be rephrased or additional experiments including Bak should be taken into account. In addition, the title suggests that the IMM permeabilization is a regulated process that involves Bax and Bak, and not purely a (mechanical?) consequence of the opening of big holes of Bax and Bak in the OMM.

3. Overall the paper is descriptive, and quantification of the observed phenomena is missing. For example, they mention that the release of Omi-cherry precedes the visual appearance of MOM pores (page 5), however this is not obvious from figures 1D and 1E. The same for the release of Omi-cherry and matrix calcein (Fig 3D). One of the main conclusions of the manuscript is that mitochondria inner membrane permeabilises after MOMP and allows the release of mtDNA to the cytosol. However, some evidence is still missing or need further clarification. They should include a graph showing the temporal difference between these events. First, it would be desirable to quantify the kinetics of the events reported in Fig. 1D (Omi release, mitochondria fragmentation and mtDNA release). Second, it would be required to correlate Omi release with matrix Calcein intensity (Fig. 4F). The timing of the events will be very useful in order to conclude that mitochondria inner membrane can permeabilize subsequent to MOMP and preceding mtDNA release.

1. The authors mention that 'growing BAX-mediated pores in the mitochondrial outer membrane enables extrusion of the mitochondrial inner membrane'. However, there is no direct evidence to reach this conclusion. Triple labelling of Bax pores, MOM and MIM would be required to address this point. In the discussion section, the authors comment about the growing of Bax pores over time as one of the conclusions of their results. However, in the present study there is a lack of quantification that limits to reach this conclusion.

Minor comments:

- Many figures miss scale bars (e.g Fig 1A,B,C ; Fig EV1D,F; Fig 3, many of the "zoomed in" figures). In Fig. 4 and Fig. EV 5, scale bar of the pictures are missing. This is extremely important in order to conclude that Bax pores grow over time, as the authors mention in the text.
- In fig 1D the color code is not mentioned as well as the zoomed area. Also the Omi mcherry is not evident, it would be worth to show a snapshot of the separated channels.
- In Fig EV 1A, there is no cell death in the presence of the caspase inhibitor QVD. However, according to other publications and to the introduction of this manuscript, if caspase activity is blocked, cell death still occurs. How can the authors explain the results shown in this graph?
- In Fig. 1E, could the authors also show the images for OMI and TFAM in the zoom in of single mitochondrion?
- Figure 1D legend, the use of transient or stable cell lines should be reviewed, since there is contradiction between main text and legend.
- In Figure 2, the authors show that MIMP allow the release of mtDNA. Additional experiments to show MIMP were performed in Figure 4E. Maybe the authors should consider to combine Figure 4E with Figure 2 to
- The references in the text do not match the labeling of the subfigures in Fig 3. (Figure 3D and F are exchanged)
- The authors show that mtDNA release occurs under caspase-inhibited conditions. However, the publication in Science shows that mtDNA release also occurs in the presence of caspases. Could the authors rule out or discuss the possibility of mtDNA release in the presence of caspases?

- A general question for the authors is: why are they using anti-DNA in many figures instead of a more specific marker for the mtDNA (such as TFAM)?
- provide all evidences of MIMP in the same figure.

Referee #2:

General comments

The current manuscript describes an elegant series of experiments using super resolution microscopy to resolve the extrusion of the mitochondrial inner membrane and matrix components (including mtDNA) into the cytoplasm likely through BAX "pores". This reveals the mechanism by which cGAS/STING signalling is triggered during apoptosis when caspases are blocked. The study is well constructed using powerful super resolution imaging, experiments well controlled and appropriately interpreted.

Using similar super resolution imaging approaches, the data presented is consistent with that reported in a very recent publication from MacArthur et al (DOI: 10.1126/science.aao6047). You could argue that this impacts the novelty of the current findings. However, these studies were clearly contemporary and such extrusion of the mitochondrial inner membrane and matrix contents challenges the dogma in the field regarding the events of apoptosis and has been questioned by leaders in the field (DOI: 10.1016/j.cmet.2015.05.019). I believe that independent validation in such situations is important. In addition, the current study shows evidence indicating that mtDNA extrudes beyond the confines of the inner membrane and so is presumably exposed to the cytoplasm, although the mechanism remains unclear.

Specific major concerns

1. The authors show elegant time lapse videos, snapshots and tomography of single cells. A valuable addition would be image analysis and quantitation of these events. This is particularly important given the timing of the analyses performed. Cells were analysed by microscopy 3h post treatment with ABT737 and ActD. However, at this time there was minimal cell death based on the Sytox analysis of cell viability in Figure EV1. Why was the 3 h time point chosen and how can the authors conclude that the cells visualised in Figure 1B had undergone MOMP when no cyt c release/ or Omi-mCherry analysis was performed in these experiments? Quantitation would also provide more convincing evidence that Bax/Bak DKO cells analysed at this timepoint did not exhibit mtDNA release.
2. The authors use a number of different models to trigger apoptosis ABT+ActD, ABT+S63845, ABT treatment of Mcl1^{-/-}. Can they provide evidence that the findings are not stimulus dependent or at least provide some explanation why the different models were used?
3. The authors should correlate mtDNA release with the triggering of the cGAS/STING pathway.
4. An interesting statement in the Discussion was that the mtDNA release occurs stochastically following MOMP. It would be a valuable addition to quantify the time between MOMP and MIMP on a per mitochondria basis. Significant variability would suggest that the eventual permeabilisation of the inner membrane is likely an unregulated event.
5. The authors should specify the exact number of independent experiments were performed rather than n=>2.

Minor

1. Abstract, "In a temporal manner, we find....over time." It is not necessary to state "In a temporal manner" as well as "over time".
2. Introduction: Suggest that authors state that it is intrinsic apoptosis that requires MOMP (first line of Intro).
3. Introduction: "...clear away dead CELL corpses..."
4. To aid the reader, describe that ActD was used to effectively inhibit the labile MCL1.
5. ABT-737 also inhibits BCL-w.
6. Ref Kotchshy et al paper for the MCL1 inhibitor.
7. Reference the original work that showed BAX/BAK oligomerisation cause MOMP (inc. Wei et al

2001).

8. "...we adapted an assay TO measure..."

9. Discussion: "...relates to our recent finding THAT sub-lethal..."

10. Does the quantitation of "%mtDNA release" in Drp1 and CypD knock-out cells refer to "%mitochondria with mtDNA released" ?

1st Revision - authors' response

16 May 2018

We are naturally pleased with both reviewers' positive evaluations and appreciate their constructive critiques. By addressing the points raised, in our opinion, this study is significantly strengthened.

Referee #1:

"I think the paper definitely deserves publication as the observations are very impressive, of broad interest, and they strengthen the findings from McArthur et al Science 2018. They come to similar conclusion than the Science paper by using comparable, though different, methods."

However there are a couple of points that should be considered:

"1. From their data they find out that the IMM gets permeabilized allowing the release of DNA nucleoids. How do they explain that the DNA doesn't diffuse at a certain point in the cytosol away from the mitochondria?"

Response: The reviewer raises an interesting point, even over prolonged periods we find that mtDNA nucleoids remain in close proximity to mitochondria (**Reviewer Figure 1**). There may be various reason for this including the large size of nucleoids (>100nm) limiting free diffusion and the restricted diffusion of DNA within the cytoplasm (Lukacs, Haggie et al., 2000).

Reviewer Figure 1: Airyscan images of U2OS cells treated with 10µM ABT-737, 2µM S63845 in the presence of 20µM qVD-OPh for 3h, 16h or 24h.

"In the Science paper they talk about IMM herniation with few permeabilization events, still enough to trigger the cGas-STING signaling pathway. How do they reconcile their findings with the ones from McArthur et al? As they do not perform live cell imaging of the IMM, could their observation be due to methodological artifacts and/or to resolution limitation? The authors should in the best case perform live cell imaging of stained IMM, or alternatively, provide a valid explanation in the discussion section to the above mentioned issue. Also very important, the authors do not provide any evidence how mtDNA comes into contact with cGAS/STING, their conclusion is speculative and should be moderated."

Response: As the reviewer suggested, in new experiments we have performed live cell imaging, simultaneously imaging the MOM (SNAP), IMM (AIF (1-90)-Scarlet) and TFAM-mClover. Similar to our fixed cell analysis (mtDNA relative to the IMM (AIF immunostaining), as expected, in healthy cells we see matrix TFAM-mClover enveloped in IMM (AIF-mScarlet) and post-MOMP we

observe extrusion of TFAM-GFP together with AIF-mScarlet, followed by separation of TFAM from AIF-mScarlet, indicative of MIMP (**Figure 4D, Movie 7**). Obvious herniation events are not visible, we think that this is most likely due to lower resolution of our Airyscan microscope relative to the lattice light sheet microscope used by McArthur, Kile and colleagues. Based on our fixed cell analysis of AIF and mtDNA, MIMP would appear more prevalent in our cells relative to ones used by McArthur et al., this suggests cell type differences in the propensity of mitochondria to undergo MIMP, potentially affecting the magnitude of cGAS-STING signaling. We now discuss this possibility (page 11). We expect that upon MIMP, cytosolic mtDNA has direct access to bind cGAS and activate STING signalling; in support, others have shown (via co-precipitation) cytosolic mtDNA binding cGAS during caspase-independent cell death (White, McArthur et al., 2014).

"2. The title of the article 'Activated Bax/Bak enable mitochondrial inner membrane permeabilisation and mtDNA release' is misleading, since there are no results about Bak in this manuscript. The title should be rephrased or additional experiments including Bak should be taken into account. In addition, the title suggests that the IMM permeabilization is a regulated process that involves Bax and Bak, and not purely a (mechanical?) consequence of the opening of big holes of Bax and Bak in the OMM."

Response: To investigate this further, we generated single BAX and BAK deficient U2OS cells to determine the contribution of BAX and BAK towards mitochondrial mtDNA release (**Figure 1F - I**). Loss of either BAX or BAK did not block mtDNA release during CICD, implying redundancy. Secondly, we generated BAX/BAK deleted and BAX or BAK deleted SVEC cells to investigate effects on cGAS/STING activation during CICD (using transcriptional upregulation of IFN- β as a readout of STING activity)(**Figure 2**). Consistent with previous data, IFN- β upregulation following BH3-mimetic treatment was dependent on MOMP (absent in BAX/BAK deleted cells), caspase inhibition and STING. Importantly, deletion of either BAX or BAK failed to inhibit IFN- β upregulation, consistent with a redundant role on mtDNA release. We agree with the referee that the title could be misconstrued as BAX/BAK directly permeabilising the inner membrane (which is not directly demonstrated) as such we have now titled it " Mitochondrial Inner Membrane Permeabilisation Enables mtDNA Release During Apoptosis"

"3. Overall the paper is descriptive, and quantification of the observed phenomena is missing. For example, they mention that the release of Omi-cherry precedes the visual appearance of MOM pores (page 5), however this is not obvious from figures 1D and 1E. The same for the release of Omi-cherry and matrix calcein (Fig 3D). One of the main conclusions of the manuscript is that mitochondria inner membrane permeabilises after MOMP and allows the release of mtDNA to the cytosol. However, some evidence is still missing or need further clarification. They should include a graph showing the temporal difference between these events. First, it would be desirable to quantify the kinetics of the events reported in Fig. 1D (Omi release, mitochondria fragmentation and mtDNA release). Second, it would be required to correlate Omi release with matrix Calcein intensity (Fig. 4F). The timing of the events will be very useful in order to conclude that mitochondria inner membrane can permeabilize subsequent to MOMP and preceding mtDNA release."

Response: We thank the reviewer for raising this important point. Accordingly, where feasible, we have now included quantification throughout the revised manuscript, including (but not limited to) the specific expts. raised by the referee (MOMP, relative to OMM widening and TFAM release and Omi release relative to loss of matrix Calcein intensity).

"1. The authors mention that 'growing BAX-mediated pores in the mitochondrial outer membrane enables extrusion of the mitochondrial inner membrane'. However, there is no direct evidence to reach this conclusion. Triple labelling of Bax pores, MOM and MIM would be required to address this point. In the discussion section, the authors comment about the growing of Bax pores over time as one of the conclusions of their results. However, in the present study there is a lack of quantification that limits to reach this conclusion."

Response: In the revised manuscript we now quantify, in BAX/BAK proficient U2OS cells, the visible MOM pores over time in individual mitochondria, this revealed a gradual widening of pores over time that preceded extrusion of TFAM (**Figure 3C and 3D**). As suggested by the reviewer, in new experiments (**Figure 6D and E**) we have imaged fixed samples for activated BAX (6A7 antibody), IMM and MOM, where we see extrusion of the inner membrane through visible MOM pores that are decorated with activated BAX, modifying our discussion to 'BAX-mediated pores in the mitochondrial outer membrane enables extrusion of the mitochondrial inner membrane'.

Minor comments:

"• Many figures miss scale bars (e.g Fig 1A,B,C ; Fig EV1D,F; Fig 3, many of the "zoomed in" figures). In Fig. 4 and Fig. EV 5, scale bar of the pictures are missing. This is extremely important in order to conclude that Bax pores grow over time, as the authors mention in the text."

Response: We apologise for the oversight, scale bars are now included throughout.

"• In fig 1D the color code is not mentioned as well as the zoomed area. Also the Omi mcherry is not evident, it would be worth to show a snapshot of the separated channels."

Response: This is now done

"• In Fig EV 1A, there is no cell death in the presence of the caspase inhibitor QVD. However, according to other publications and to the introduction of this manuscript, if caspase activity is blocked, cell death still occurs. How can the authors explain the results shown in this graph?"

Response: In the short-term caspase inhibition protects cells post-MOMP from dying, however over a longer-period cells die regardless. Consistent with this paradigm, in a new expt. (**Figure 1B**) assaying cell death by clonogenic survival assay, we find that caspase inhibition fails to allow clonogenic survival post-MOMP (stimulated by BH3-mimetics).

"• In Fig. 1E, could the authors also show the images for OMI and TFAM in the zoom in of single mitochondrion?"

Response: This is now done (**Figure 3C**).

"• Figure 1D legend, the use of transient or stable cell lines should be reviewed, since there is contradiction between main text and legend."

Response: This is now done

"• In Figure 2, the authors show that MIMP allow the release of mtDNA. Additional experiments to show MIMP were performed in Figure 4E. Maybe the authors should consider to combine Figure 4E with Figure 2 to"

"• provide all evidences of MIMP in the same figure."

Response: We appreciate the suggestion, but are of the opinion, with the presentation of data investigating BAX activity relative to MIMP (in **Figure 4 original ms. now Figure 6 in revised**), the logical narrative would be to include it with the rest of the BAX data rather than move it to an earlier figure.

"• *The references in the text do not match the labeling of the subfigures in Fig 3. (Figure 3D and F are exchanged)"*

Response: Apologies for the oversight, now corrected.

"• *The authors show that mtDNA release occurs under caspase-inhibited conditions. However, the publication in Science shows that mtDNA release also occurs in the presence of caspases. Could the authors rule out or discuss the possibility of mtDNA release in the presence of caspases?"*

Response: While we could not detect mtDNA release during caspase-dependent apoptosis, this clearly can occur as McArthur and colleagues have shown. In new expts. (**Figure 2D**) we do observe a modest upregulation of IFN- β mRNA even during mitochondrial apoptosis, suggestive of MIMP; we suspect the inability to visualise this release is due to rapid cell rounding during (caspase-dependent) apoptosis. This is incorporated into our discussion (page 11), whereby we speculate that mtDNA release (and potentially cGAS-STING activity) may be more likely in cell types with lower levels of caspase activity.

"• *A general question for the authors is: why are they using anti-DNA in many figures instead of a more specific marker for the mtDNA (such as TFAM)?"*

Response: Given the role of mtDNA in activating cGAS-STING signalling, we primarily focused on detecting mtDNA during CICD. In new experiments we have co-stained for DNA, TFAM and MOM (**Figure 3A**). Validating the utility of the anti-DNA antibody as a means of detecting mtDNA, we find strong co-localisation between TFAM and DNA signal both before and after treatment (BH3-mimetic/QVD), in the latter both co-localising beyond the MOM.

Referee #2

General comments

The current manuscript describes an elegant series of experiments using super resolution microscopy to resolve the extrusion of the mitochondrial inner membrane and matrix components (including mtDNA) into the cytoplasm likely through BAX "pores". This reveals the mechanism by which cGAS/STING signalling is triggered during apoptosis when caspases are blocked. The study is well constructed using powerful super resolution imaging, experiments well controlled and appropriately interpreted.

Using similar super resolution imaging approaches, the data presented is consistent with that reported in a very recent publication from MacArthur et al (DOI: 10.1126/science.aao6047). You could argue that this impacts the novelty of the current findings. However, these studies were clearly contemporary and such extrusion of the mitochondrial inner membrane and matrix contents challenges the dogma in the field regarding the events of apoptosis and has been questioned by leaders in the field (DOI: 10.1016/j.cmet.2015.05.019). I believe that independent validation in such situations is important. In addition, the current study shows evidence indicating that mtDNA

extrudes beyond the confines of the inner membrane and so is presumably exposed to the cytoplasm, although the mechanism remains unclear.

Specific major concerns

"1. The authors show elegant time lapse videos, snapshots and tomography of single cells. A valuable addition would be image analysis and quantitation of these events. This is particularly important given the timing of the analyses performed. Cells were analysed by microscopy 3h post treatment with ABT737 and ActD. However, at this time there was minimal cell death based on the Sytox analysis of cell viability in Figure EV1. Why was the 3 h time point chosen and how can the authors conclude that the cells visualised in Figure 1B had undergone MOMP when no cyt c release/ or Omi-mCherry analysis was performed in these experiments? Quantitation would also provide more convincing evidence that Bax/Bak DKO cells analysed at this timepoint did not exhibit mtDNA release. "

Response: We thank the reviewer for raising these points. In the revised manuscript we now provide extensive quantitation throughout. Regarding the specific point raised (whether the cells had undergone MOMP or not at 3h post-ABT737/ActD treatment), in matched samples we have monitored MOMP by immunostaining for mitochondrial release of cytochrome c. This typically shows over >90% cells have undergone MOMP at this 3h time point. This data is referred to in the discussion and now included as **Figure EV1C and D**. In new expts. (**Figure 1I**) we have quantified mtDNA release in Bax/Bak deleted cells, as well as single Bax and Bak deleted cells; this shows that Bax/Bak deleted cells are completely inhibited in their ability to release mtDNA, unlike Bax or Bak deleted cells.

"2. The authors use a number of different models to trigger apoptosis ABT+ActD, ABT+S63845, ABT treatment of Mcl1-/. Can they provide evidence that the findings are not stimulus dependent or at least provide some explanation why the different models were used?"

Response: The necessity to use various treatments was based on the following: Pro-longed live-cell imaging post-MOMP with ActD/BH3-mimetic was not possible due to phototoxicity (due to ActD). Second, being a transcriptional inhibitor, Act D was incompatible with analysis of cGAS-STING transcriptional activation. Prior to our acquisition of the Mcl-1 inhibitor S63845, Mcl-1 deletion (via CRISPR/Cas9) was used to achieve rapid responses to ABT-737, this has been largely usurped by availability of the Mcl-1 inhibitor. In all cases, treatments were chosen to engage mitochondrial apoptosis in a rapid, synchronous manner since this greatly facilitates subsequent microscopy. Taking this into account, our ability to detect mtDNA release is independent of stimulus applied and is detectable in all cell types tested so far.

"3. The authors should correlate mtDNA release with the triggering of the cGAS/STING pathway."

Response: In new experiments we have directly investigated this. Here we used SVEC cells (which have intact cGAS/STING signaling) where we generated lines, via CRISPR/Cas9 genome editing deficient in STING, BAX/BAK, BAX or BAK. Cells were stimulated to undergo CICD and analysed by qPCR for IFN- β and by microscopy for mtDNA release (**Figure 2**). Under conditions of CICD, as previously reported, IFN is upregulated in a STING and BAX/BAK dependent manner. At a similar timepoint, mitochondrial mtDNA release was also observed, correlating mtDNA release with cGAS/STING activation.

"4. An interesting statement in the Discussion was that the mtDNA release occurs stochastically following MOMP. It would be a valuable addition to quantify the time between MOMP and MIMP on a per mitochondria basis. Significant variability would suggest that the eventual permeabilisation of the inner membrane is likely an unregulated event."

Response: We thank the reviewer for raising this point and have quantified accordingly (**Figure EV3D**). Indeed, we see a wide variability in the time between MOMP and MIMP (assayed by TFAM release) and some mitochondria fail to undergo MIMP following MOMP, suggestive that it may be an unregulated event.

5. The authors should specify the exact number of independent experiments were performed rather than $n \geq 2$.

Response: We have now defined the exact amount of independent experiments.

Minor

1. Abstract, "In a temporal manner, we find...over time." It is not necessary to state "In a temporal manner" as well as "over time".

Response: Modified as suggested.

2. Introduction: Suggest that authors state that it is intrinsic apoptosis that requires MOMP (first line of Intro).

Response: Modified as suggested.

3. Introduction: "...clear away dead CELL corpses..."

Response: Modified as suggested.

4. To aid the reader, describe that ActD was used to effectively inhibit the labile MCL1.

Response: Modified as suggested.

5. ABT-737 also inhibits BCL-w.

Response: Modified as suggested.

6. Ref Koteschy et al paper for the MCL1 inhibitor.

Response: Added in revised version

7. Reference the original work that showed BAX/BAK oligomerisation cause MOMP (inc. Wei et al 2001).

Response: Added in revised version

8. "...we adapted an assay TO measure..."

Response: Added in revised version

9. Discussion: "...relates to our recent finding THAT sub-lethal..."

Response: Added in revised version

10. Does the quantitation of "%mtDNA release" in Drp1 and CypD knock-out cells refer to "%mitochondria with mtDNA released" ?

Response: refers to amount of mtDNA release per cell, now clarified

References

Lukacs GL, Haggie P, Seksek O, Lechardeur D, Freedman N, Verkman AS (2000) Size-dependent DNA mobility in cytoplasm and nucleus. *J Biol Chem* 275: 1625-9

White MJ, McArthur K, Metcalf D, Lane RM, Cambier JC, Herold MJ, van Delft MF, Bedoui S, Lessene G, Ritchie ME, Huang DC, Kile BT (2014) Apoptotic caspases suppress mtDNA-induced STING-mediated type I IFN production. *Cell* 159: 1549-62

2nd Editorial Decision

6 June 2018

Thank you for submitting a revised version of your manuscript. Your study has been seen by the two original referees and we have now received their comments, which are enclosed below for your information.

As you can see, both referees are fully satisfied with the new data and referee#1 suggests that you cite one additional reference in the text. However, before we can go ahead and officially accept your manuscript for publication there are a few editorial issues concerning text and figures that I would ask you to address in a final revised version.

REFEREE REPORTS

Referee #1:

The authors have addressed the reviewers concerns and further supported the permeabilization of the IMM with additional evidence.

Minor comments:

-The authors may consider to add a reference where Bleicken et al, *JBC* 2013 provide evidence for the growing size of Bax and Bak pores.

Referee #2:

The authors have adequately addressed all of my comment.

2nd Revision - authors' response

11 June 2018-06-15

Referee #1:

“The authors have addressed the reviewers concerns and further supported the permeabilization of the IMM with additional evidence.

Minor comments:

*-The authors may consider to add a reference where Bleicken et al, *JBC* 2013 provide evidence for the growing size of Bax and Bak pores.”*

Response: We agree with the referee that this study is relevant to our discussion and now discuss the suggested reference, in the context of our findings, on page 11.

Corresponding Author Name: Stephen Tait

Journal Submitted to: The EMBO Journal

Manuscript Number: EMBOJ-2018-99238